# Directed Lithiation of Protected 4-Chloropyrrolopyrimidine: Addition to Aldehydes and Ketones Aided by Bis(2-dimethylaminoethyl)ether

**DOI:** 10.3390/molecules28030932

**Published:** 2023-01-17

**Authors:** Frithjof Bjørnstad, Eirik Sundby, Bård Helge Hoff

**Affiliations:** 1Department of Chemistry, Norwegian University of Science and Technology (NTNU), NO-7491 Trondheim, Norway; 2Department of Material Science, Norwegian University of Science and Technology (NTNU), NO-7491 Trondheim, Norway

**Keywords:** directed lithiation, pyrrolopyridimine, bis(2-dimethylaminoethyl) ether, SEM-protection, lithiation

## Abstract

Pyrrolopyrimidines are important scaffolds for the preparation of bioactive molecules. Therefore, developing efficient and flexible ways for selective functionalization of the pyrrolopyrimidine skeleton is of interest. We have investigated lithiation-addition at C-6 of protected 4-chloro-7*H*-pyrrolo [2,3-*d*]pyrimidine as a route to new building blocks for medicinal chemistry. It was found that bis(2-dimethylaminoethyl) ether as an additive increased the yield in the additional reaction with benzaldehyde. Deuterium oxide quench experiments showed that this additive offered both a higher degree of lithiation and increased stability of the lithiated intermediate. The substrate scope of the protocol was investigated with 16 aldehydes and ketones, revealing the method to be excellently suited for reaction with aldehydes, cyclohexanone derivatives and 2,2,2-trifluoroacetophenone, while being less efficient for acetophenones. Yields in the range of 46–93% were obtained.

## 1. Introduction

Due to their bioisosteric relationship with purines, pyrrolopyrimidines are attractive pharmacophores in medicinal chemistry [1,2], with a wide range of applications, including antiviral, antimicrobial, inflammatory, and cancer indications. The chemistry pertaining to a variety of advanced pyrrolopyrimidines has been reviewed [3,4]. To prepare new pyrrolopyrimidines of interest, having access to advanced building blocks with accessible and selective reactive handles is of major importance. One key starting material for pyrrolopyrimidine synthesis is 4-chloro-7*H*-pyrrolo[2,3-*d*]pyrimidine (Figure 1). The C-4 position is reactive in both nucleophilic aromatic substitutions (S_N_Ar) and cross-couplings, while N-7 can be alkylated, arylated, or glycosylated. Additional reactive handles for cross-coupling chemistry can be inserted by electrophilic aromatic substitution at C-5, resulting in structure **I** [5,6], and the N-7 benzyl analogue of **II** has been borylated using iridium-catalysis to provide **III** [7]. Further, introducing a coordinating protection group at N-7 allows for directed lithiation at C-6, providing us with the reactive intermediate **IV**. Quenching of **IV** with iodine provides the building block **V** [8,9], and the use of CO_2_ gives the carboxylic acid **VI** [10]. Another possibility is to react the lithiated intermediate **IV** with aldehydes and ketones to result in structures like **VII**. Sakomoto et al. used *n*-butyllithium (*n*-BuLi) to functionalize a benzenesulfonyl protected 2,4-dimethoxy-pyrrolo[2,3-*d*]pyrimidine [11].

Zhao et al. [12] employed the same protection group and reacted **IV** with cyclic ketones, resulting in yields in the range of 55–77%. In ongoing projects to prepare bioactive pyrrolopyrimidines, we wanted to functionalize the C-6 position with a sp^2^-sp^3^ carbon linker, such as that exemplified in structure **VII**. We realized that directed lithiation, followed by quenching with ketones and aldehydes, could be a viable approach. Although we have previously found the benzenesulfonyl group useful in directing lithiation at C-6, its lability under amination conditions at C-4 is highly inconvenient [9]. Thus, the 2-(trimethylsilyl)ethoxymethyl (SEM) protection group was seen as an interesting alternative. Herein, we report our study on the directed lithiation of SEM-protected pyrrolopyrimidines and addition to aldehydes and ketones for the preparation of advanced intermediates for medicinal chemistry. We found that the lithiation process could be improved by utilizing bis(2-dimethylaminoethyl) ether as an additive, resulting in higher overall yields. Furthermore, the substrate scope has been explored by reaction with 16 different ketones and aldehydes of varying acidity and coordinating groups.

## 2. Results and Discussion

### 2.1. Study of the Lithiation Process

We have previously used SEM-protected pyrrolopyrimidine **1** in lithiation-iodination at C-6 [8], and knew that the directing properties of the SEM group was suitable. As a starting point, we treated **1** with LDA at −78 °C in THF, followed by quenching with benzaldehyde (**2a**) to give the product **3a**, see Figure 2. This resulted in an isolated yield of 68%. Higher consumption of the starting material was observed when the reaction mixture was allowed to reach room temperature, but with no increase in product formation. Instead, the by-products **4** and **6** were formed (Figure 2), resulting from the nucleophilic attack by diisopropylamine at C-4. The use of lithium bis(trimethylsilyl)amide resulted in no conversion and appears too weak to deprotonate this scaffold, while *n*-BuLi produced a complex mixture with reaction also at C-4. To establish a foundation to improve the process, we first decided to evaluate the lithiation step in more detail. To monitor conversion of **1** to the lithiated intermediate **IV**, samples of the reaction mixture after lithiation were quenched in D_2_O giving the C-6 deuterium analogue **5**. The ratio between the D-6 and H-6 analogues could be estimated by ^1^H NMR analysis. This ratio was used to estimate the degree of lithiation (% Li). Initially, the reproducibility of the measured conversion seemed low, and did not correspond with yields obtained in iodination. Challenges with this type of quench have previously been noted [13,14]. Seebach concluded that the lithiated intermediate likely forms an aggregate with diisopropylamine, which, due to its proximity, efficiently competes with added D_2_O in the quench [14]. Tuning of the sample preparation by introducing a sonication step with D_2_O gave an average conversion of 83 ± 2% (*n* = 5) in the lithiation process at −78 °C. The accuracy of this measurement is dependent on the quality of the integration by ^1^H NMR spectroscopy.

A key question was the stability of the lithiated intermediate **IV** under operating conditions. It has been reported that deprotonations of this type can be reversible in nature, especially at non-cryogenic temperatures [15,16]. Therefore, we conducted experiments at three different temperatures (−78 °C, −40 °C and −10 °C) and monitored the degree of lithiation as a function of time (Figure 1). At −10 °C, the degree of lithiation was low, even after 1 h, and at both −10 °C and −40 °C the degree of lithiation decreased rapidly as a function of time, whereas this process was considerably slower at −78 °C. Additionally, elevated levels of the by-product **4** were noted at higher temperatures. To further improve upon the reaction, we evaluated the effects of LDA amount, solvent composition, and additives on the conversion to the lithiated intermediate **IV**. The reaction was quenched with D_2_O after 1 h reaction time. The results of these experiments are displayed in Table 1. Using 1.1 equivalents (equiv.) of LDA resulted in lower conversion (Table 1, entry 1) than seen when using the initial conditions 1.6 equiv. (entry 2). Employing 2 equiv. of LDA (entry 3) had a slight positive effect. To evaluate the effect of solvent polarity on the lithiation, reactions were also conducted in toluene and mixtures of toluene and THF. Lowering the polarity of the reaction medium had a detrimental effect on lithiation (entries 4–7), highlighting the need for a polar coordinating solvent. Lithium chloride is known to increase the speed of the lithiation step [15,17]. In our case, we noticed a marginal positive effect using 0.1 and 0.5 equiv. of added LiCl (entries 8–9), and no effect when using 1 equiv. (entry 10). However, LiCl is highly hygroscopic, and its use requires careful handling. Another group of additives used to improve directed metalation are chelating agents; although their role in lithiations can be quite complex [18], they are often claimed to prevent aggregation of LDA and the lithiated intermediates. The most commonly used chelator, *N*,*N*,*N*′,*N*′-tetramethylethylenediamine (TMEDA), had no clear effect on the lithiation (entry 11), while an obvious positive effect was seen when employing bis[2-(*N*,*N*-dimethylamino)ethyl] ether (BDMAE), where the degree of lithiation increased to an average of 96% (entry 12). BDMAE appears to be somewhat overlooked as an additive in lithiations, but it has previously been investigated in combination with LDA in dehydrobromination of a bromoalkene [19], in Grignard reactions [20], and in lithiation of *N*-tosyl indols [21].

With these initial positive results, we went on to evaluate the stability of the lithium complex at −78 °C as a function of time in the presence of BDMAE. Importantly, the stability was found to be higher than that seen in lithiation without the additive (see Figure 1). Furthermore, this naturally improved conversion in the addition step and, in a preparative experiment with benzaldehyde, increased the isolated yield from 68 to 93%.

### 2.2. Substrate Scope

With these promising results, we went on to evaluate the substrate scope of the lithiation-addition protocol using BDMAE as an additive, see Figure 3. Different classes of aldehydes and ketones were included, having a difference in reactivity and acidity. The results from these reactions are displayed in Table 2.

After lithiation, the success of the process was monitored by ^1^H NMR (Li %), and the efficiency of the addition step was checked by a ^1^H NMR measurement of levels of **1** and product **3**. The lithiation process proceeded well in all cases, with >94% measured lithiation. Careful development of isolation protocols for each derivative was not performed; thus, several of the compounds had to be purified twice. Surprisingly, the removal of the starting material was challenging in several cases. First, the lithiation-addition reactions with 6 aldehydes were tested (entries 1–6). The conversion in the addition step was excellent in all cases, and the reaction with the enolizable aldehyde **2f** (entry 6) also proceeded well. Majewski reported that when benzaldehydes were treated with LDA, generated from *n*-BuLi and diisopropylamine, reduction to the corresponding benzyl alcohols occurred [22]. These by-products were not seen in our experiments. The acetophenones were a more challenging substrate class. The more electron-rich ketones (entries 7–8) with higher pKa [23], appeared to be more sluggish in the addition reaction than the other acetophenones (entries 9–11). One hypothesis that can explain the mediocre conversion in this series is that the acetophenones equilibrate with lithiated **IV** and are trapped as enolates. Aldol condensations could also be envisioned. The complete formation of the enolate of acetophenone by LDA deprotonation (THF, −78 °C) has previously been observed in trapping experiments with trimethylsilyl chloride by Silva et al. [24]. However, in self-condensation experiments between this enolate and acetophenone, only 11% of the aldol product was formed. We did not detect aldol condensation products in our experiments. Moreover, we also found no aldol products when *p*-methoxyacetophenone (**2g**) was deprotonated by LDA followed by quenching with **2g**. Wu et al. [21] also observed mediocre conversion between 2-lithioindols and acetophenone and proposed that the lithium enolate of acetophenone was stabilized by the chelating agent, rendering it unreactive at low temperatures. Having no acidic protons and a highly electrophilic carbonyl carbon [25], 2,2,2-trifluoroacetophenone (**2l**) was a less complicated substrate. Full conversion was seen for this substrate (entry 12). Finally, we evaluated four cyclic ketones in the lithiation-addition protocol. Only minor differences in conversion were noted for the 6-membered ring ketones (entries 13–15); however, the protocol seems excellently suited for this substrate class overall. Surprisingly, the reaction with cyclopentanone only resulted in 72% conversion; ^1^H NMR of the crude material, in addition to 3p, mostly contained **1** and cyclopentanone. There were no indications of aldol products being formed. The altered reactivity going from six- to five-membered rings might be due to internal strain differences as observed in other addition reactions [26,27].

The directed lithiation-addition protocol reported here is complementary to cross-coupling methodology, as sp^2^-sp^3^ C-C bonds are formed. In other cases, directed lithiation-addition can be an alternative to cross-coupling, which is most obvious for the cyclic ketones. For instance, the product **3m,** obtained in 91% of yields, is a simple dehydration away from the Suzuki–Miyaura product **8,** obtained in 83% of yields from **7**, see Figure 4. The dehydration can be affected under acidic conditions typically used to deprotect the SEM group, or by mesylation of the alcohol and treatment with a sterically-hindered base, as reported by Chaitanya et al.[28]). Since very few alkenyl boronic acids are commercially available, and at high prices, this addition protocol can be highly useful in these settings.

All of these transformations can be further tuned both in terms of details in the addition protocol and the purification method. As pointed out by Collum [18], the role of additives such as TMEDA is not necessarily as a lithium chelating agent. Thus, it can be that certain substates perform best with other additives. The use of BDMAE in combination with LDA seems excellently suited for the addition of lithiated pyrrolopyrimidines to aldehydes, trifluoroketones and cyclohexanone derivatives.

## 3. Materials and Methods

### 3.1. Chemicals and Analysis

All solvents and reagents used in the project were purchased from VWR and Merck. 4-Chloro-7*H*-pyrrolo[2,3-*d*]pyrimidine was obtained from 1 Click Chem, while compound **7** was prepared as previously described [8]. Silica gel chromatography was performed using silica gel 60A, purchased from VWR with a pore size of 40–63 µm. Solvents were dried on a Braun MB SPS-800 Solvent Purification System (MBRAUN, Garching, Germany) and stored over molecular sieves (4 Å) for 24 h prior to use. ^1^H- and ^13^C-NMR spectra were recorded using a Bruker Advance III HD NMR spectrometer (Bruker, Billerica, MA, US) with a Smartprobe 5 mm probe head, operating at 400 MHz or 600 MHz, and carbon spectra at 100 MHz or 150 MHz, respectively. Samples were mainly analyzed in DMSO-*d*_6_ or chloroform-*d* where specified. ^1^H and ^13^C NMR chemical shifts are in ppm relative to the DMSO solvent peak at 2.50 ppm and 39.5 ppm, respectively. The NMR spectra are included in the Appendix A). High-resolution mass spectroscopy (HRMS) was performed using a WaterTM’s Synapt G2-S Q-TOF instrument (Waters, Milford, MA, USA). Samples were ionized by Electrospray Ionization (ESI/70eV) and analyzed using an Atmospheric Solids Analysis Probe (ASAP). Calculated exact mass and spectra processing was performed by WatersTM Software (Masslynx V4.1 SCN871).

### 3.2. Degree of Lithiation (Li %)

A sample (ca 0.1 mL) was removed from the reaction mixture with a dry glass Hamiltonian syringe. The sample was directly added to a vial containing D_2_O (0.5 mL). The vial was capped and sonicated at 22 °C for 30 min. The vial was removed from the sonication bath and vigorously stirred for 1 min. The vial was left at 22 °C for an additional 30 min before being extracted with CH_2_Cl_2_ (1mL). The solvent was then removed under pressure, and the sample was added DMSO-*d*_6_ (0.5 mL) for ^1^H NMR analysis. Present lithiation (%Li) was estimated by employing the integrals of the H-6 and H-2 protons. The substitution of H-6 for deuterium does not result in a large enough shift difference at H-2 to differentiate the H-6 and D-6 analogs. However, the shift belonging to H-6 at 7.22, resulting from the residual non-lithiated substrate, can be used. The integral belonging to H-6 is compared to the signal arising from H-2, originating from both the H-6 and D-6 compounds. This allows for quick and easy estimation of % lithiation using the formula: Li % = 100 × [∫(H-2)–∫(H-6)]/∫(H-2). Here, ∫(H-2) is the total integral of the H-2 protons (both analogues), and ∫(H-6) is the integral of the doublet at 7.87 originating from the H-6 analogue.

### 3.3. Synthesis

#### 3.3.1. General Procedure A: Directed Lithiation without Additives

Under an N_2_ atmosphere 4-chloro-7-((2-(trimethylsilyl) ethoxy)methyl)-7*H*-pyrrolo[2,3-*d*]pyrimidine (500 mg, 1.83 mmol) was dissolved in dry THF (4 mL) and cooled down to −78 °C. Then, LDA (2 M in THF/*n*-hexane/ethylbenzene, 1.47 mL, 2.93 mmol, 1.6 equiv) was added dropwise over 30 min by cannulation. This was followed by the dropwise addition of the ketone/aldehyde (2.19 mmol, 1.2 equiv.) dissolved in THF (2 mL). After another 60 min, the reaction mixture was quenched with saturated NH_4_Cl solution (0.5 mL) and stirred until ambient temperature was reached. The mixture was concentrated and CH_2_Cl_2_ (25 mL) and water (30 mL) were added. After phase separation, the water phase was extracted with more CH_2_Cl_2_ (2 × 20 mL) and washed with brine (20 mL). The combined organic phase was dried over Na_2_SO_4_ and the solvent was removed under reduced pressure. The crude product was purified by silica gel flash chromatography, as specified.

#### 3.3.2. General Procedure B: Directed Lithiation Using Bis(N,N′-dimethylaminoethyl) ether as Additive

Under an N_2_ atmosphere 4-chloro-7-((2-(trimethylsilyl) ethoxy)methyl)-7*H*-pyrrolo[2,3-*d*]pyrimidine (500 mg, 1.83 mmol) was dissolved in dry THF (4 mL) and cooled down to −78 °C. Bis(N,N’-dimethylaminoethyl) ether (2.75 mmol, 1.5 equiv) was added through the septum, followed by the addition of LDA (2 M in THF/n-hexane/ethylbenzene) (1.47 mL, 2.93 mmol, 1.6 equiv) dropwise over 30 min by cannulation. This was followed by the dropwise addition of the ketone/aldehyde (2.19 mmol, 1.2 equiv.) dissolved in THF (2 mL). After another 60 min, the reaction mixture was quenched with saturated NH_4_Cl solution (0.5 mL) and stirred until the ambient temperature was reached. The mixture was concentrated and CH_2_Cl_2_ (25 mL) and water (30 mL) were added. After phase separation, the water phase was extracted with more CH_2_Cl_2_ (2 × 20 mL) and washed with brine (20 mL). The combined organic phase was dried over Na_2_SO_4_ and the solvent was removed under reduced pressure. The crude product was purified by silica gel flash chromatography, as specified.

#### 3.3.3. 4-Chloro-7-((2-(trimethylsilyl)ethoxy)methyl)-7*H*-pyrrolo[2,3-*d*]pyrimidine (**1**)

Dry DMF (50 mL) was cooled to 0 °C and added to NaH (60% dispersion in oil) (2.7 g, 80.4 mmol). Next, 4-chloro-7*H*-pyrrolo[2,3-*d*]pyrimidine (10.2 g, 67.1 mmol) was dissolved in dry DMF (20 mL) and added portion-wise to the chilled suspension over 15 min. The reaction mixture was left stirring at 0 °C for 20 min. Then, SEM-Cl (14.2 mL, 80.4 mmol) was added. The mixture was left stirring for 1 h, while cooling, before quenching with sat. aq. NH_4_Cl (2 mL). The mixture was transferred to a round-bottom flask and concentrated in a vacuum. The concentrated residue was partitioned between CH_2_Cl_2_ (40 mL) and water (50 mL). The layers were separated, and the water-phase was extracted with more CH_2_Cl_2_ (3 × 50 mL). The combined organic layers were washed with water (4 × 50 mL) and brine (50 mL), dried with anhydrous Na_2_SO_4_, filtered and concentrated in vacuo. The residue was purified by column chromatography on silica gel (*n*-pentane/EtOAc 10:1, R_f_ = 0.24) yielding 14.3 g (50.3 mmol, 76%) of a clear oil. ^1^H NMR (400 MHz, DMSO-*d_6_*) 8.68 (s, 1H), 7.87 (d, J = 3.6 Hz, 1H), 6.71 (d, J = 3.6 Hz, 1H), 5.65 (s, 2H), 3.52 (t, J = 8.0 Hz, 2H), 0.82 (t, J = 8.0 Hz, 2H), −0.10 (s, 9H). ^13^C NMR (100 MHz, DMSO-*d*_6_) δ: 151.3, 150.8, 150.7, 131.5, 116.9, 99.3, 72.9, 65.8, 17.1, −1.5 (3C); IR (neat, cm^−1^): 3120 (br,w), 3088 (w), 2950 (m), 2896 (w), 1587 (s), 1542 (s), 1348 (s), 1033 (s), 833 (s), 744 (s); HRMS (ES+, *m*/*z*): found 284.099, calcd. C_12_H_19_N_3_OSiCl [M + H]^+^, 284.0986.

#### 3.3.4. (4-Chloro-7-((2-(trimethylsilyl)ethoxy)methyl)-7*H*-pyrrolo[2,3-*d*]pyrimidin-6-yl)(phenyl)methanol (**3a**)

Compound **1** (650 mg, 2.28 mmol) and benzaldehyde (0.291 mL, 2.74 mmol) were reacted as described in general procedure B. The reaction time was 1 h. Purification by silica gel chromatography (gradient from *n*-pentane/acetone/MeOH, 90:10:2 to 85:15:2, TLC: *n*-pentane/acetone/MeOH, 90:10:2, R_f_ = 0.21), produced 835 mg (2.14 mmol, 93%) of a thick, oil. ^1^H NMR (600 MHz, DMSO-*d_6_*) δ 8.66 (s, 1H), 7.43–7.34 (m, 5H), 6.39 (d, *J* = 5.5 Hz, 1H), 6.27 (d, *J* = 0.9 Hz, 1H), 6.05 (d, *J* = 5.5 Hz, 1H), for N-CH_2_-O an AB-system: δ_A_= 5.77, δ_B_ = 5.56, J_AB_ =11.0 Hz, 3.50–3.37 (m, 2H), 0.83–0.70 (m, 2H), −0.11 (s, 9H); ^13^C NMR (151 MHz, DMSO-*d_6_*) δ 152.5, 150.7, 150.3, 146.4, 141.5, 128.3 (2C), 127.9, 126.9 (2C), 115.9, 98.1, 70.7, 67.6, 65.5, 17.0, −1.47 (3C). IR (neat, cm^−1^): 3483 (br, w), 3057 (w), 2950 (m), 2896 (w), 1557 (s), 1455 (s), 1252 (s), 1163 (s), 834 (s), 763 (s); HRMS (ES+, *m*/*z*): found 390.1407, calcd for C_19_H_25_ClN_3_O_2_Si, [M + H]^+^, 390.1404.

#### 3.3.5. (4-Chloro-7-((2-(trimethylsilyl)ethoxy)methyl)-7*H*-pyrrolo[2,3-*d*]pyrimidin-6-yl)(4-methoxyphenyl)methanol (**3b**)

Compound **1** (456 mg, 1.61 mmol) and 4-methoxybenzaldehyde (0.231 mL, 1.92 mmol) were reacted as described in general procedure B. The reaction time was 1 h. Purification by silica gel chromatography (gradient from *n*-pentane/EtOAc, 9:1 to 8:2, TLC: *n*-pentane/EtOAc, 10:1, R_f_ = 0.42), produced 546 mg (1.34 mmol, 83%) of a clear oil. ^1^H NMR (600 MHz, DMSO-*d_6_*) δ 8.66 (s, 1H), 7.35–7.30 (m, 2H), 6.98–6.93 (m, 2H), 6.31 (s, 1H), 6.28 (d, *J* = 5.4 Hz, 1H), 5.99 (d, *J* = 5.3 Hz, 1H), for N-CH_2_-O an AB-system: δ_A_= 5.76, δ_B_ = 5.50, J_AB_ =10.8 Hz, 3.76 (s, 3H), 3.49–3.36 (m, 2H), 0.83–0.72 (m, 2H), -0.10 (s, 9H); ^13^C NMR (151 MHz, DMSO-*d_6_*) δ 158.9, 152.6, 150.7, 150.2, 146.8, 133.5, 128.3 (2C), 115.9, 113.7 (2C), 97.9, 70.7, 67.3, 65.5, 55.1, 17.03, −1.46 (3C). IR (neat, cm^−1^): 3308 (br, w), 2955 (m), 1587 (s), 1510 (s), 1352 (s), 1243 (s), 1078 (s), 917 (s), 815 (s); HRMS (ES+, *m*/*z*): found 420.1512, calcd for C_20_H_27_ClN_3_O_3_Si, [M + H]^+^, 420.1510.

#### 3.3.6. (4-Chloro-7-((2-(trimethylsilyl)ethoxy)methyl)-7*H*-pyrrolo[2,3-*d*]pyrimidin-6-yl)(2-fluorophenyl)methanol (**3c**)

Compound **1** (531 mg, 1.87 mmol) and 2-fluorobenzaldehyde (0.238 mL, 2.24 mmol) were reacted as described in general procedure B. The reaction time was 1 h. Purification by silica gel chromatography (*n*-pentane/acetone/MeOH, 90:10:2, R_f_ = 0.27) produced 588 mg (1.44 mmol, 78%) of a light red oil. ^1^H NMR (400 MHz, DMSO-*d_6_*) δ 8.68 (s, 1H), 7.56 (td, *J* = 7.6, 1.9 Hz, 1H), 7.47–7.36 (m, 1H), 7.31–7.20 (m, 2H), 6.50 (d, *J* = 5.9 Hz, 1H), 6.35 (d, *J* = 5.9 Hz, 1H), 6.18 (d, *J* = 0.9 Hz, 1H), for N-CH_2_-O an AB-system: δ_A_= 5.79, δ_B_ = 5.65, J_AB_ =11.0 Hz, 3.51–3.34 (m, 2H), 0.89–0.62 (m, 2H). −0.16 (s, 9H); ^13^C NMR (101 MHz, DMSO-*d_6_*) δ 159.4 (d, *J_CF_* = 245 Hz), 152.6, 150.9, 150.4, 145.1, 130.1 (d, *J_CF_* = 8.2 Hz), 128.6 (d, *J_CF_* = 3.4 Hz), 128.4 (d, *J_CF_* = 13.7 Hz), 124.6 (d, *J_CF_* = 3.4 Hz), 115.8, 115.3 (d, *J_CF_* = 21.6 Hz), 98.1, 70.8, 65.5, 61.1,17.0, −1.51 (3C); ^19^F NMR (565 MHz, DMSO-*d_6_*, C_6_F_6_) d: −121.1; IR (neat, cm^−1^): 3342 (br, w), 2951 (m), 1589 (s), 1455 (s), 1347 (s), 1246 (s), 1197 (s), 1077 (s), 831 (s), 748 (s); HRMS (ES+, *m*/*z*): found 408.1312, calcd for C_19_H_24_ClFN_3_O_2_Si, [M + H]^+^, 408.1310.

#### 3.3.7. (4-Chloro-7-((2-(trimethylsilyl)ethoxy)methyl)-7*H*-pyrrolo[2,3-*d*]pyrimidin-6-yl)(thiophen-2-yl)methanol (**3d**)

Compound **1** (491 mg, 1.73 mmol) and thiophene-2-carbaldehyde (0.193 mL, 2.07 mmol) were reacted as described in general procedure B. The reaction time was 1 h. Purification by silica gel chromatography twice (first: gradient from *n*-pentane/EtOAc, 90:10 to 85:15, TLC: *n*-pentane/EtOAc, 10:1, R_f_ = 0.40, then *n*-pentane/acetone/(MeOH 85:5:2, TLC: *n*-pentane/acetone/MeOH, 85:5:2, Rf = 0.36) produced 468 mg (1.18 mmol, 70%) of a yellow wax. ^1^H NMR (400 MHz, DMSO-*d_6_*) δ 8.67 (s, 1H), 7.54 (dd, *J* = 5.0, 1.3 Hz, 1H), 7.08–6.99 (m, 2H), 6.73 (d, *J* = 5.5 Hz, 1H), 6.49 (d, *J* = 0.8 Hz, 1H), 6.32 (d, *J* = 5.5 Hz, 1H), for N-CH_2_-O an AB-system: δ_A_= 5.75, δ_B_ = 5.56, J_AB_ =11.0 Hz, 3.52–3.36 (m, 2H), 0.84–0.72 (m, 2H), −0.11 (s, 9H);^13^C NMR (101 MHz, DMSO-*d_6_*) δ 152.5, 150.9, 150.5, 145.8, 145.6, 126.8, 126.0, 125.4, 115.9, 97.7, 70.7, 65.6, 63.62, 17.0, −1.47 (3C); IR (neat, cm^−1^): 3427 (br, w), 2953 (m), 1587 (s), 1538 (s), 1348 (s), 1246 (s), 1064 (s), 830 (s), 695 (s); HRMS (ES+, *m*/*z*): found 396.0971, calcd for C_17_H_23_ClN_3_O_2_SSi, [M + H]^+^, 396.0968.

#### 3.3.8. (4-Chloro-7-((2-(trimethylsilyl)ethoxy)methyl)-7*H*-pyrrolo[2,3-*d*]pyrimidin-6-yl)(pyridin-2-yl)methanol (**3e**)

Compound **1** (463 mg, 1.63 mmol) and picolinaldehyde (0.187 mL, 1.95 mmol) were reacted as described in general procedure B. The reaction time was 1 h. Purification by silica gel chromatography twice (first gradient from *n*-pentane/EtOAc/MeOH, 86:13:1 to 82:17:1, TLC: *n*-pentane/EtOAc/MeOH, 82:17:1, R_f_ = 0.40, then *n*-pentane/acetone 95:5 to 90:10, TLC: *n*-pentane/acetone/, 85:5, Rf = 0.24), produced 476 mg (1.22 mmol, 75%) of a yellow oil. ^1^H NMR (600 MHz, DMSO-*d_6_*) δ 8.65 (s, 1H), 8.50 (dd, *J* = 4.9, 1.8, 1H), 7.87 (td, *J* = 7.7, 1.8 Hz, 1H), 7.66 (dd, *J* = 7.9, 1.2 Hz, 1H), 7.34 (dd, *J* = 7.6, 4.8 Hz, 1H), 6.62 (d, *J* = 5.6 Hz, 1H), 6.33 (d, *J* = 0.8 Hz, 1H), 6.12 (d, *J* = 5.6 Hz, 1H), 5.79 (s, 2H), 3.43 (t, *J* = 8.1 Hz, 2H), 0.85–0.72 (m, 2H), −0.10 (s, 9H);^13^C NMR (151 MHz, DMSO-*d_6_*) δ 160.9, 152.4, 150.7, 150.3, 148.5, 145.8, 137.1, 123,0, 121.0, 116.0, 97.9, 70.8, 69.1, 65.5, 17.0, −1.47 (3C); IR (neat, cm^−1^): 3361 (br, w), 2921(m), 1587 (m), 1073 (s), 832 (s); HRMS (ES+, *m*/*z*): found 391.1360, calcd for C_18_H_24_ClN_4_O_2_Si, [M + H]^+^, 391.1357.

#### 3.3.9. (4-Chloro-7-((2-(trimethylsilyl)ethoxy)methyl)-7*H*-pyrrolo[2,3-*d*]pyrimidin-6-yl)(cyclohexyl)methanol (**3f**)

Compound **1** (510 mg, 1.81 mmol) and cyclohexanecarbaldehyde (0.27 mL, 2.2 mmol) were reacted as described in general procedure B. The reaction time was 1 h. Purification by silica gel chromatography (gradient from *n*-pentane/EtOAc, 9:1 to 8:2, TLC: *n*-pentane/EtOAc, 7:1, R_f_ = 0.37), produced 537 mg (1.35 mmol, 75%) of a clear oil. ^1^H NMR (600 MHz, DMSO-*d_6_*) δ 8.64 (s, 1H), 6.57 (s, 1H), for N-CH_2_-O an AB-system: δ_A_= 5.75, δ_B_ = 5.71, J_AB_ =11.0 Hz, 5.54 (d, *J* = 5.5 Hz, 1H), 4.59 (dd, *J* = 7.6, 5.5 Hz, 1H), 3.50 (t, *J* = 8.1 Hz, 2H), 1.97–1.85 (m, 2H), 1.75–1.58 (m, 4H), 1.38–1.31 (m, 1H), 1.26–0.99 (m, 4H), 0.89–0.77 (m, 2H), −0.09 (s, 9H); ^13^C NMR (151 MHz, DMSO-*d_6_*) δ 152.4, 150.2, 149.8, 146.4, 116.1, 97.4, 70.4, 70.20, 65.6, 42.1, 29.5, 28.2, 26.0, 25.5 (2C), 17.2, −1.5 (3C); IR (neat, cm^−1^): 3344 (br, w), 2920 (s), 2950 (m), 2850 (w), 1587 (s), 1449 (s), 1351(s), 1248 (s), 1200 (s), 1075 (s), 833 (s), 741 (s); HRMS (ES+, *m*/*z*): found 396.1878, calcd for C_19_H_31_ClN_3_O_2_Si, [M + H]^+^, 396.1874.

#### 3.3.10. 1-(4-Chloro-7-((2-(trimethylsilyl)ethoxy)methyl)-7*H*-pyrrolo[2,3-*d*]pyrimidin-6-yl)-1-(4-methoxyphenyl)ethan-1-ol (**3g**)

Compound **1** (340 mg, 1.19 mmol) and 4-methoxyacetophenone (215 mg, 1.43 mmol) were reacted as described in general procedure B. The reaction time was 1 h. Purification by silica gel chromatography was performed twice (first gradient from *n*-pentane/acetone 95:5 to 90:10, then *n*-pentane/acetone/MeOH, 85:15:2, TLC: *n*-pentane/acetone, 15:1, R_f_ = 0.30), producing 241 mg (0.555 mmol, 46%) of a yellow oil. ^1^H NMR (400 MHz, DMSO-*d_6_*) δ 8.63 (s, 1H), 7.31–7.22 (m, 2H), 6.90–6.81 (m, 2H), 6.72 (s, 1H), 6.20 (s, 1H), 5.47 (s, 2H), 3.72 (s, 3H), 3.26–3.10 (m, 2H), 1.90 (s, 3H), 0.57–0.50 (m, 2H), −0.13 (s, 9H); ^13^C NMR (101 MHz, DMSO-*d_6_*) δ 158.1, 153.1, 150.7, 150.2, 148.8, 138.1, 126.2 (2C), 115.8, 113.3 (2C), 97.7, 71.9, 71.4, 65.2, 54.9, 31.8, 17.2, −1.54 (3C); IR (neat, cm^−1^): 3337 (br, w), 2947 (m), 1675 (m), 1588 (s),1511 (s), 1352 (s),1244 (s), 1081 (s), 833 (s); HRMS (ES+, *m*/*z*): found 434.1671, calcd for C_21_H_29_ClN_3_O_3_Si, [M + H]^+^, 434.1666.

#### 3.3.11. 1-(4-(Benzyloxy)phenyl)-1-(4-chloro-7-((2-(trimethylsilyl)ethoxy)methyl)-7*H*-pyrrolo[2,3-*d*]pyrimidin-6-yl)ethan-1-ol (**3h**)

Compound **1** (313 mg, 1.10 mmol) and 1-(4-(benzyloxy)phenyl)ethan-1-one (298 mg, 1.32 mmol) were reacted as described in general procedure B. The reaction time was 1 h. Purification by silica gel chromatography was performed twice (first gradient from *n*-pentane/acetone 95:5 to 90:10, then *n*-pentane/acetone/MeOH, 90:10:2, (TLC: 90:10:2, R_f_ = 0.40) produced 181 mg (0.36 mmol, 32%) of a white wax. ^1^H NMR (400 MHz, DMSO-*d_6_*) δ 8.63 (s, 1H), 7.44–7.34 (m, 5H), 7.29–7.24 (m, 2H), 6.98–6.91 (m, 2H), 6.72 (s, 1H), 6.21 (s, 1H), 5.48 (s, 2H), 5.05 (s, 2H), 3.27–3.12 (m, 2H), 1.90 (s, 3H), 0.59–0.52 (m, 2H), −0.13 (s, 9H). ^13^C NMR (101 MHz, DMSO *d_6_*) δ 157.3, 153.1, 150.8, 150.3, 148.8, 138.4, 137.1, 128.4 (2C), 127.80, 127.6 (2C), 126.3 (2C), 115.8, 114.2 (2C), 97.7, 71.9, 71.4, 69.2, 65.2, 31.8, 17.2, −1.50 (3C); IR (neat, cm^−1^): 3419 (br, w), 2952 (m), 2902 (m), 1674 (s), 1588 (s), 1506 (s), 1245 (s), 1169 (s), 827 (s), 748 (s), 707 (s); HRMS (ES+, *m*/*z*): found 510.1981, calcd for C_27_H_33_ClN_3_O_3_Si, [M + H]^+^, 510.1979.

#### 3.3.12. 1-(4-Chloro-7-((2-(trimethylsilyl)ethoxy)methyl)-7*H*-pyrrolo[2,3-*d*]pyrimidin-6-yl)-1-phenylethan-1-ol (**3i**)

Compound **1** (520 mg, 1.83 mmol) and acetophenone (0.256 mL, 2.19 mmol) were reacted as described in general procedure B. The reaction time was 1 h. Purification by silica gel chromatography (*n*-pentane/acetone, 10:1, R_f_ = 0.32) produced 455 mg (1.17 mmol, 61%) of a pale, thick oil. ^1^H NMR (400 MHz, DMSO-*d_6_*) δ 8.64 (s, 1H), 7.41–7.31 (m, 2H), 7.32–7.26 (m, 2H), 7.26–7.18 (m, 1H), 6.77 (s, 1H), 6.29 (s, 1H), for N-CH_2_-O an AB-system: δ_A_= 5.49, δ_B_ = 5.44, J_AB_ =11.0 Hz, 3.22–3.07 (m, 2H), 1.92 (s, 3H), 0.57–0.46 (m, 2H), −0.12 (s, 9H); ^13^C NMR (101 MHz, DMSO-*d_6_*) δ 153.1, 150.8, 150.3, 148.5, 146.17, 128.0 (2C), 126.8, 124.9 (2C), 115.8, 97.9, 72.2, 71.4, 65.1, 31.9, 17.1, −1.48 (3C); IR (neat, cm^−1^): 3413 (br, w), 2951 (m), 2885 (m), 1587 (s), 1347 (s), 1248 (s), 1248 (s), 1068(s), 833 (s), 741 (s), 696 (s); HRMS (ES+, *m*/*z*): found 404.1568, calcd for C_20_H_27_ClN_3_O_2_Si, [M + H]^+^, 404.1561.

#### 3.3.13. 1-(4-Chloro-7-((2-(trimethylsilyl)ethoxy)methyl)-7*H*-pyrrolo[2,3-*d*]pyrimidin-6-yl)-1-(4-(trifluoromethyl)phenyl)ethan-1-ol (**3j**)

Compound **1** (510 mg, 1.70 mmol) and 1-(4-(trifluoromethyl)phenyl)ethan-1-one (402 mg, 2.14 mmol) were reacted as described in general procedure B. The reaction time was 1 h. Purification by silica gel chromatography twice (first: gradient *n*-pentane/acetone, 95:5 to 90:10, then *n*-pentane/acetone/CH_2_Cl_2_, 85:15:2, TLC: *n*-pentane/acetone, 20:1, R_f_ = 0.28), produced 391 mg (0.83 mmol, 46%) of a clear oil. ^1^H NMR (400 MHz, DMSO-*d_6_*) δ 8.65 (s, 1H), 7.64 (d, *J* = 8.3 Hz, 2H), 7.57 (d, *J* = 8.2 Hz, 2H), 6.92 (s, 1H), 6.59 (s, 1H), 5.58 (d, *J* = 9.9 Hz, 1H), 5.52 (d, *J* = 9.9 Hz, 1H), 3.07–2.92 (m, 2H), 1.94 (s, 3H), 0.35–0.18 (m, 2H), −1.70 (9H); ^13^C NMR (101 MHz, DMSO-*d_6_*) δ 153.1, 151.0 (2C), 150.5, 147.4, 127.3 (q, *J_CF_* = 32 Hz), 125.7 (2C), 125.3 (q, *J_CF_* = 190 Hz), 124.9 (q, *J_CF_* = 3.8 Hz, 2C), 115.7, 98.3, 72.2, 71.3, 64.7, 32.0, 16.8, −1.7 (3C); ^19^F NMR (565 MHz, DMSO-*d_6_*, C_6_F_6_) d: −63.0; IR (neat, cm^−1^): 3357 (br, w), 2953 (m), 1590 (m), 1322 (s), 1159 (s), 1120 (s), 1092 (s), 832 (s); HRMS (ES+, *m*/*z*): found 472.1437, calcd for C_21_H_26_ClF_3_N_3_O_2_Si, [M + H]^+^, 472.1434.

#### 3.3.14. 1-(4-Chloro-7-((2-(trimethylsilyl)ethoxy)methyl)-7*H*-pyrrolo[2,3-*d*]pyrimidin-6-yl)-1-(4-nitrophenyl)ethan-1-ol (**3k**)

Compound **1** (490 mg, 1.72 mmol) and 4-nitroacetophenone (340 mg, 2.06 mmol) were reacted as described in general procedure B. The reaction time was 1 h. Purification by silica-gel chromatography (gradient *n*-pentane/acetone, 95:5 to 90:10, TLC: *n*-pentane/acetone, 10:1, R_f_ = 0.31), produced 373 mg (0.83 mmol, 49%) of a white wax. ^1^H NMR (400 MHz, DMSO-*d_6_*) δ 8.67 (s, 1H), 8.18–8.14 (m, 2H), 7.67–7.59 (m, 2H), 6.95 (s, 1H), 6.72 (s, 1H), for N-CH_2_-O an AB-system: δ_A_= 5.55, δ_B_ = 5.52, J_AB_ =10.0 Hz, 3.00 (dd, *J* = 9.2, 7.5 Hz, 2H), 1.96 (s, 3H), 0.25 (td, *J* = 7.8, 1.7 Hz, 2H), −0.20 (s, 9H); ^13^C NMR (101 MHz, DMSO-*d_6_*) δ 154.0, 153.1, 151.1, 150.7, 146.9, 146.3, 126.4 (2C), 123.2 (2C), 115.7, 98.5, 72.2, 71.2, 64.72, 31.8, 16.9, −1.7 (3C); IR (neat, cm^−1^): 3312 (br, w), 2951 (m), 1518 (s), 1344 (s), 1075 (s), 836 (s), 1696 (s); HRMS (ES+, *m*/*z*): found 449.1412, calcd for C_20_H_26_ClN_4_O_4_Si, [M + H]^+^, 449.1411.

#### 3.3.15. 1-(4-Chloro-7-((2-(trimethylsilyl)ethoxy)methyl)-7*H*-pyrrolo[2,3-*d*]pyrimidin-6-yl)-2,2,2-trifluoro-1-phenylethan-1-ol (**3l**)

Compound **1** (341 mg, 1.20 mmol) and 2,2,2-trifluoro-1-phenylethan-1-one (0.132 mL, 1.44 mmol) were reacted as described in general procedure B. The reaction time was 1 h. Purification by silica gel chromatography (gradient n-pentane/EtOAc, 10:1 to 5:1, TLC: *n*-pentane/EtOAc, 10:1, R_f_ = 0.31) produced 381 mg (0.87 mmol, 73%) of a yellow oil. ^1^H NMR (400 MHz, DMSO-*d_6_*) δ 8.74 (s, 1H), 8.08 (s, 1H), 7.46–7.33 (m, 5H), 6.90–6.85 (m, 1H), 5.48–5.37 (m, 2H), 3.06–2.89 (m, 2H), 0.39–0.37 (m, 2H), −0.15 (s, 9H); ^13^C NMR (101 MHz, DMSO-*d_6_*) δ 152.8, 151.8, 151.4, 138.2, 135.8, 128.9, 128.1 (2C), 127.7 (q, J_CF_ = 280 Hz), 127.1 (2C), 115.4, 99.8, 75.9 (q, J_CF_= 29 Hz), 71.5, 65.1, 16.9, −1.52 (3C); ^19^F NMR (565 MHz, DMSO-*d_6_*, C_6_F_6_) d: −77.6; IR (neat, cm^−1^): 3484 (br, w), 3057 (w), 1557 (m), 1359 (m), 1163 (s), 1069 (s), 1252 (s), 994 (s), 864 (s); HRMS (ES+, *m*/*z*): found 458.1287, calcd for C_20_H_24_ClF_3_N_3_O_2_Si, [M + H]^+^, 458.1278.

#### 3.3.16. 1-(4-Chloro-7-((2-(trimethylsilyl)ethoxy)methyl)-7*H*-pyrrolo[2,3-*d*]pyrimidin-6-yl)cyclohexan-1-ol (**3m**)

Compound **1** (346 mg, 1.21 mmol) and cyclohexanone (0.151 mL, 1.46 mmol) were reacted as described in general procedure B. The reaction time was 1 h. Purification by silica gel chromatography (*n*-pentane/EtOAc, 10:1, R_f_ = 0.44) produced 423 mg (1.11 mmol, 91%) of white wax. ^1^H NMR (600 MHz, DMSO-*d_6_*) δ: 8.64 (s, 1H), 6.54 (s, 1H), 5.99 (s, 2H), 5.28 (s, 1H), 3.64 (t, J = 8.0 Hz, 2H), 2.11–2.17 (m, 2H), 1.79–1.86 (m, 2H), 1.70–1.79 (m, 2H), 1.59–1.64 (m, 1H), 1.48–1.54 (m, 2H), 1.22–1.31 (m, 1H), 0.83 (t, J = 8.0 Hz, 2H), −0.09 (s, 9H); ^13^C NMR (150 MHz, DMSO-*d_6_*) δ: 153.2, 150.7, 150.5, 150.0, 115.8, 96.1, 71.6, 69.5, 66.1, 36.9 (2C), 25.1, 21.3 (2C), 17.4, −1.5 (3C); IR (neat, cm^−1^): 3364 (br, w), 3152 (w), 2938 (m), 2847 (w), 1586 (m), 1547 (m); HRMS (ES+, *m/z*): found 382.1721 calcd for C_18_H_29_N_3_O_2_SiCl, [M + H]^+^, 382.1718.

#### 3.3.17. 1-(4-Chloro-7-((2-(trimethylsilyl)ethoxy)methyl)-7*H*-pyrrolo[2,3-*d*]pyrimidin-6-yl)-4-methylcyclohexan-1-ol (**3n**)

Compound **1** (546 mg, 1.92 mmol) and 4-methylcyclohexan-1-one (0.283 mL, 2.31 mmol) were reacted as described in general procedure B. The reaction time was 1 h. Purification by silica gel chromatography (*n*-pentane/EtOAc, 7:1, R_f_ = 0.63) produced 594 mg (1.50 mmol, 79%) of a slightly yellow wax. ^1^H NMR (400 MHz, DMSO-*d_6_*) δ 8.66 (s, 1H), 6.62 (s, 1H), 5.94 (s, 2H), 5.40 (s, 1H), 3.71–3.59 (m, 2H), 2.47–2.33 (m, 2H), 1.82–1.74 (m, 4H), 1.64–1.60 (1, 2H), 1.18–1.05 (m, 2H), 0.89 (d, *J* = 6.7 Hz, 3H), 0.85–0.81 (m, 2H), −0.08 (s, 9H); ^13^C NMR (101 MHz, DMSO-*d_6_*) δ 153.2, 150.6, 150.0, 148.3, 115.7, 98.0, 71.4, 70.0, 66.1, 36.0 (2C), 30.5 (2C), 29.8, 20.3, 17.41, −1.44 (3C); IR (neat, cm^−1^): 3444 (br, w), 2946 (m), 1715 (m), 1245 (m), 1092 (s), 831 (s), 747 (s); HRMS (ES+, *m/z*): found 396.1881, calcd for C_19_H_31_ClN_3_O_2_Si, [M + H]^+^, 396.1874.

#### 3.3.18. 4-(4-Chloro-7-((2-(trimethylsilyl)ethoxy)methyl)-7*H*-pyrrolo[2,3-*d*]pyrimidin-6-yl)tetrahydro-2H-pyran-4-ol (**3o**)

Compound **1** (542 mg, 1.91 mmol) and tetrahydro-4*H*-pyran-4-one (0.283 mL, 2.29 mmol) were reacted as described in general procedure B. The reaction time was 1 h. Purification by silica gel chromatography (*n*-pentane/EtOAc, 2:1, R_f_ = 0.40) yielding 524 mg (1.37 mmol, 72%) of a clear yellow oil.^1^H NMR (600 MHz, DMSO-*d_6_*) δ 8.67 (s, 1H), 6.61 (s, 1H), 5.99 (s, 2H), 5.63 (s, 1H), 3.84–3.77 (m, 2H), 3.75–3.68 (m, 2H), 3.65–3.58 (m, 2H), 2.12–2.05 (m, 4H), 0.86–0.81 (m, 2H), -0.09 (s, 9H); ^13^C NMR (151 MHz, DMSO-*d_6_*) δ 153.3, 150.7, 150.3, 149.3, 115.7, 96.5, 71.4, 67.2, 66.0, 62.5 (2C), 37.2 (2C), 17.4, −1.5 (3C); ^13^C NMR (151 MHz, DMSO) δ 153.3, 150.7, 150.3, 149.3, 115.7, 96.5, 71.4, 67.2, 66.0, 62.5 (2C), 37.2 (2C), 17.4, −1.47 (3C); IR (neat, cm^−1^): 3482 (br, w), 3380 (m), 3057 (m), 1558 (m), 1360 (s), 1162 (s), 1069 (s), 967(s), 835 (s); HRMS (ES+, *m/z*): found 384.1514, calcd for C_17_H_27_ClN_3_O_3_Si, [M + H]+, 384.1510.

#### 3.3.19. 1-(4-Chloro-7-((2-(trimethylsilyl)ethoxy)methyl)-7*H*-pyrrolo[2,3-*d*]pyrimidin-6-yl)cyclopentan-1-ol (**3p**)

Compound **1** (463 mg, 1.63 mmol) and cyclopentanone (0.174 mL, 1.95 mmol) were reacted as described in general procedure B. The reaction time was 1 h. Purification by silica gel chromatography (*n*-pentane/EtOAc, 7:1, R_f_ = 0.51) produced 301 mg (0.81 mmol, 51%) of a clear oil. ^1^H NMR (600 MHz, DMSO-*d_6_*) δ 8.65 (s, 1H), 6.58 (s, 1H), 5.91 (s, 2H), 5.35 (s, 1H), 3.64–3.58 (m, 2H), 2.11–2.04 (m, *J* = 5.6, 3.2 Hz, 5H), 1.90–1.80 (m, 1H), 1.75–1.65 (m, 2H), 0.87–0.79 (m, 2H), −0.09 (s, 9H); ^13^C NMR (151 MHz, DMSO-*d_6_*) δ 153.3, 150.5, 150.1, 148.7, 115.8, 95.9, 78.0, 71.3, 65.9, 40.1 (2C), 23.00 (2C), 17.4, −1.46 (3C); IR (neat, cm^−1^): 3431 (br, w), 2955 (m), 1587 (m), 1549 (w), 1354 (m), 1063 (s), 830 (s), 771 (s); HRMS (ES+, *m/z*): found 368.1566, calcd for C_17_H_27_ClN_3_O_2_Si, [M + H]^+^, 368.1561.

#### 3.3.20. *N*,*N*-Diisopropyl-7-((2-(trimethylsilyl)ethoxy)methyl)-7*H*-pyrrolo[2,3-*d*]pyrimidin-4-amine (**4**)

Compound **4** was isolated after a reaction of **1** using general procedure A, allowing the temperature to react after the addition of LDA, followed by 15 h of reaction time. Purification by silica gel column chromatography (*n*-pentane/EtOAc, 7:1, *R_f_* = 0.41) produced 152 mg (0.436 mmol, 25%) of compound **4**. ^1^H NMR (600 MHz, DMSO-*d_6_*) δ: 8.21 (s, 1H), 6.62 (d, J = 3.4 Hz, 1H), 6.19 (d, J = 3.4 Hz, 1H), 5.15 (s, 2H), 4.56 (hept, J = 6.7 Hz, 1H), 3.73 (hept, J = 6.7 Hz, 1H), 3.49–3.41 (m, 2H), 1.24–1.21 (m, 12H), 0.84–0.78 (m, 2H), −0.07 (s, 9H); ^13^C NMR (101 MHz, DMSO-*d_6_*) δ 152.1, 150.5, 119.1, 117.4, 108.6, 74.0, 72.9, 65.2, 46.6, 45.6, 23.4 (2C), 19.4 (2C), 17.2, −1.5 (3C). HRMS (ES+, *m/z*): found 349.2426, calcd for C_18_H_33_N_4_OSi, [M + H]+, 349.2423.

#### 3.3.21. 4-Chloro-6-deutero-7-((2-(trimethylsilyl)ethoxy)methyl)-7*H*-pyrrolo[2,3-*d*]pyrimidine (**5**)

Under an N_2_ atmosphere 4-chloro-7-((2-(trimethylsilyl)ethoxy)methyl)-7*H*-pyrrolo[2,3-*d*]pyrimidine (64 mg, 0.226 mmol) was dissolved in dry THF (2 mL) and cooled down to −78 °C. Bis(*N*,*N*’-dimethylaminoethyl) ether (0.064 mL 0.338 mmol, 1.5 equiv) was added through the septum, followed by the addition of LDA (2 M in THF/*n*-hexane/ethylbenzene) (0.180 mL, 0.36 mmol, 1.6 equiv) dropwise over 30 min by cannulation. This was followed by the dropwise addition of deuterium oxide (1.2 mL) and THF (1 mL). The reaction mixture was allowed to warm to room temperature and stirred for 30 min before being sonicated for an additional 30 min at room temperature. The reaction mixture was quenched with saturated NH_4_Cl solution (0.5 mL). Before being concentrated and added CH_2_Cl_2_ (25 mL) and water (30 mL). After the phase separation, the water phase was extracted with more CH_2_Cl_2_ (2 × 20 mL) and washed with brine (20 mL). The combined organic phase was dried over Na_2_SO_4,_ and the solvent was removed under reduced pressure. The residue was purified by column chromatography on silica gel (*n*-pentane/EtOAc, 10:1, R_f_ = 0.24) yielding 54 mg (0.189 mmol, 84%) of a clear oil. ^1^H NMR (400 MHz, DMSO-d_6_) δ 8.68 (s, 1H), 6.71 (s, 1H), 5.65 (s, 2H), 3.54–3.51 (m, 2H), 0.86–0.83 (m,2H), −0.10 (s, 9H); ^13^C NMR (100 MHz, DMSO-d6) δ 151.3, 150.8, 150.7, 131.3 (t, J = 30.0 Hz), 116.9, 99.2, 72.9, 65.8, 59 17.1, −1.5 (3C); HRMS (ES+, *m/z*): found 285.1049, calcd. C_12_H_18_DClN_3_OSi [M + H]_+_, 285.1047.

#### 3.3.22. (4-(Diisopropylamino)-7-((2-(trimethylsilyl)ethoxy)methyl)-7*H*-pyrrolo[2,3-*d*]pyrimidin-6-yl)(phenyl)methanol (**6**)

Compound **6** was isolated after the reaction of **1,** using general procedure A and allowing the temperature to react after the addition of LDA, followed by 15 h of reaction time. Purification by silica gel column chromatography (*n*-pentane/EtOAc, 7:1, *R_f_* = 0.21) produced 74 mg (0.163 mmol, 9%), ^1^H NMR (400 MHz, DMSO–*d_6_*) δ 8.18 (s, 1H), 7.37–7.25 (m, 5H), 5.78 (d, J = 5.4 Hz, 1H), 5.71 (d, J = 5.4 Hz, 1H), 5.55 (s, 1H), for N-CH_2_-O an AB-system: δ_A_= 5.40, δ_B_ = 5.22, J_AB_ =10.6 Hz, 4.56 (hept, J = 6.7 Hz, 1H), 3.73 (hept, J = 6.7 Hz, 1H), 3.47–3.43 (m, 2H), 1.31–1.25 (m, 12H), 0.85–0.76 (m, 2H), −0.05 (s, 9H); ^13^C NMR (151 MHz, DMSO–*d_6_*) δ 152.2, 151.6, 142.7, 132.2, 127.9 (2C), 127.2, 126.6 (2C), 119.0, 107.9, 72.8, 70.4, 67.1, 65.1, 46.8, 45.8, 23.4 (2C), 19.4 (2C), 17.3, −1.4 (3C); HRMS (ES+, *m/z*): found 455.2838, calcd for C_25_H_39_N_4_O2Si, [M + H]+, 455.28.

#### 3.3.23. 4-Chloro-6-(cyclohex-1-en-1-yl)-7-((2-(trimethylsilyl)ethoxy)methyl)-7*H*-pyrrolo[2,3-*d*]pyrimidine (**8**)

4-Chloro-6-iodo-7-((2-(trimethylsilyl)ethoxy)methyl)-7*H*-pyrrolo[2,3-*d*]pyrimidine (569 mg, 1.38 mmol), cyclohex-1-en-1-ylboronic acid (238 mg, 1.89 mmol), K_2_CO_3_ (574 mg, 4.16 mmol) and Pd(dppf)Cl_2_ (51.2 mg, 69.7 μmol) were added to a Schlenk tube in an N_2_ atmosphere. Degassed H_2_O (5 mL) and 1,4-dioxane (10 mL) were added, and the reaction was stirred at 80 °C for 60 min. The vessel was then cooled to room temperature before the addition of H_2_O (20 mL) and CH_2_Cl_2_ (20 mL). After phase separation, the aqueous phase was extracted with more CH_2_Cl_2_ (3 × 20 mL) and the combined organic layers were washed with brine (20 mL) before being dried over Na_2_SO_4_, filtered and concentrated in vacuo. Purification by silica gel chromatography (*n*-pentane/EtOAc, 10:1, R_f_ = 0.32) produced 419 mg (1.15 mmol, 83%) of a clear oil. ^1^H NMR (400 MHz, DMSO-d_6_) δ 8.64 (s, 1H), 6.61 (s, 1H), 6.43 (td, J = 3.9, 1.9 Hz, 1H), 5.60 (s, 2H), 3.63–3.61 (m, 2H), 2.44–2.36 (m, 2H), 2.28–2.19 (m, 2H), 1.79–1.69 (m, 2H), 1.73–1.59 (m, 2H), 0.90–0.80 (m, 2H), −0.09 (s, 9H); ^13^C NMR (101 MHz, DMSO-*d_6_*) δ 152.9, 150.4, 149.8, 144.9, 131.8, 127.5, 116.6, 96.9, 71.2, 66.1, 27.9, 25.2, 22.2, 21.2, 17.2, −1.5 (3C); HRMS (ES+, *m/z*): found 364.1612, calcd for C_18_H_27_ClN_3_OSi, [M + H]+, 364.1611.

## 4. Conclusions

A route to new pyrrolopyrimidine building blocks using lithiation-addition reactions has been investigated. The lithiation was directed to the C-6 position by installing a 2-trimethylsilyl)ethoxymethyl group at *N*-7 of the pyrrolpyrimidine. Key to improving the process was a robust analysis of the lithiation step, in which ultrasonic treatment during D_2_O quench, followed by ^1^H NMR analysis, proved useful. A study of the lithiation process showed that increased lithiation could be achieved by including BDMAE as an additive, while LiCl had only minor effects. The role of BDMAE in the reaction is not clear, however, its presence improved conversion, storage stability and yield in the following reaction with benzaldehyde. A substrate scope study revealed that the protocol was excellently suited for lithiation-addition to aldehydes, trifluoroketones, and cyclohexanone derivatives, while more mediocre conversions and yields were obtained for acetophenones and cyclopentanone. The lithiation-addition approach for pyrrolopyrimidines complements and is also, in certain cases, an alternative to cross-coupling methodology.

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
