# Peer review of "Directed Lithiation of Protected 4-Chloropyrrolopyrimidine: Addition to Aldehydes and Ketones Aided by Bis(2-dimethylaminoethyl)ether"

_molecules, 2023, doi:10.3390/molecules28030932_

Round 1
Reviewer 1 Report
In this manuscript entitled: “Directed lithiation of protected 4-chloropyrrolopyrimidine: addition to aldehydes and ketones aided by bis(2-dimethylaminoethyl)ether”, the authors: Frithjof Bjørnstad, Eirik Sundby, Bård Helge Hoff * present the synthesis of several 7H-pyrrolo[2,3-d]pyrimidine by directed lithiation at C-6 of SEM-protected 4-chloro-7H-pyrrolo[2,3-d]pyrimidine followed by addition of the lithiated intermediate to aldehydes or ketones.
I believe that the paper may be suitable for publication in the MDPI journal Molecules after addressing the following considerations listed below.
- I suggest including other recent references.
- All headings should be written using the title case, e.g.
- in line 215, I suggest to be written ”3.1. Chemicals and Analysis” instead of “3.1 Chemicals and analysis”;
- in line 251, I suggest writing “3.3.1. General Procedure A: Directed Lithiation without Additives” instead of “3.3.1 General procedure A: directed lithiation without additives.”.
- A space is not necessary between two successive or different references, e.g. in line 26.
- In line 76, is necessary that “1H NMR analysis” be corrected to “1H NMR analysis”.
- In line 114, “N,N,N′,N′-tetramethyl ethylenediamine (TMEDA)” I suggest writing ”N,N,N′,N′-tetramethylethylenediamine (TMEDA)”.
- I suggest that the empty lines be deleted, for example, lines 161-163.
- In line 225, “1H and 13C NNR” is necessary to be corrected (“1H and 13C NMR”).
- In section 3.1, I suggest mentioning the name of the manufacturing company, country, and city of manufacture of each apparatus.
- In line 217, I suggest that “4-Chloro-7H-pyrrolo[2,3-d] pyrimidine” to be written “4-Chloro-7H-pyrrolo[2,3-d]pyrimidine”.
- In line 218, it is necessary that “preaviusly” be corrected.
- In line 219, please verify “pore size 40-63 um”.
- In lines 252, 253, 267, 268, 543, and 544, “4-chloro-7-((2-(trimethylsilyl)-ethoxy)methyl)-7H-pyrrolo[2,3-d]pyrimidine” I suggest to be written “4-chloro-7-((2-(trimethylsilyl)ethoxy)methyl)-7H-pyrrolo[2,3-d]pyrimidine”.
- In lines 265, 266, 269, and 545, “bis(N,N’- dimethylaminoethyl) ether” I suggest to be written “bis[2-(N,N-dimethylamino)ethyl] ether”.
- In line 417, “C27H32ClN3O3Si” is necessary to be corrected to “C27H33ClN3O3Si”.
- In line 538, “C18H32N4OSi,” is necessary to be corrected to “C18H33N4OSi”.
- In lines 541 and 542, “4-Chloro-6-deutro-7-((2-(trimethylsilyl)ethoxy)methyl)-7H-pyrrolo[2,3-d]pyrimidine (5)” is necessary to be written “4-Chloro-6-deutero-7-((2-(trimethylsilyl)ethoxy)methyl)-7H-pyrrolo[2,3-d]pyrimidine (5)”.
- In line 543, “N2” is necessary to be written “N2” and in line 558, “13C NMR” is necessary to be written “13C NMR”.
- In line 560, “C12H17DClN3OSi” is necessary to be corrected to “C12H18DClN3OSi”.
- In line 591, “C18H26ClN3OSi” is necessary to be corrected to “C18H27ClN3OSi”.
Author Response
- I suggest including other recent references.
Response: We have included a new reference nr 28. We think that the other references are relevant and have not changed these.
- All headings should be written using the title case, e.g.
Response: modified accordingly
- in line 215, I suggest to be written ”3.1. Chemicals and Analysis” instead of “3.1 Chemicals and analysis”;
Response: modified accordingly.
- in line 251, I suggest writing “3.3.1. General Procedure A: Directed Lithiation without Additives” instead of “3.3.1 General procedure A: directed lithiation without additives.”.
Response: modified accordingly.
- A space is not necessary between two successive or different references, e.g. in line 26.
Response: The space is inserted by the Endnote program. I hope the typesetters can adjust this.
- In line 76, is necessary that “1H NMR analysis” be corrected to “1H NMR analysis”.
Response: corrected.
- In line 114, “N,N,N′,N′-tetramethyl ethylenediamine (TMEDA)” I suggest writing ”N,N,N′,N′-tetramethylethylenediamine (TMEDA)”.
Response: corrected.
- I suggest that the empty lines be deleted, for example, lines 161-163.
Response: corrected.
- In line 225, “1H and 13C NNR” is necessary to be corrected (“1H and 13C NMR”).
Response: corrected
- In section 3.1, I suggest mentioning the name of the manufacturing company, country, and city of manufacture of each apparatus.
Response: the information is inserted.
- In line 217, I suggest that “4-Chloro-7H-pyrrolo[2,3-d] pyrimidine” to be written “4-Chloro-7H-pyrrolo[2,3-d]pyrimidine”.
Response: corrected.
- In line 218, it is necessary that “preaviusly” be corrected.
Response: corrected.
- In line 219, please verify “pore size 40-63 um”.
Response: corrected.
- In lines 252, 253, 267, 268, 543, and 544, “4-chloro-7-((2-(trimethylsilyl)-ethoxy)methyl)-7H-pyrrolo[2,3-d]pyrimidine” I suggest to be written “4-chloro-7-((2-(trimethylsilyl)ethoxy)methyl)-7H-pyrrolo[2,3-d]pyrimidine”.
Response: corrected.
- In lines 265, 266, 269, and 545, “bis(N,N’- dimethylaminoethyl) ether” I suggest to be written “bis[2-(N,N-dimethylamino)ethyl] ether”.
Response: corrected.
- In line 417, “C27H32ClN3O3Si” is necessary to be corrected to “C27H33ClN3O3Si”.
Response: corrected.
- In line 538, “C18H32N4OSi,” is necessary to be corrected to “C18H33N4OSi”.
Response: corrected.
- In lines 541 and 542, “4-Chloro-6-deutro-7-((2-(trimethylsilyl)ethoxy)methyl)-7H-pyrrolo[2,3-d]pyrimidine (5)” is necessary to be written “4-Chloro-6-deutero-7-((2-(trimethylsilyl)ethoxy)methyl)-7H-pyrrolo[2,3-d]pyrimidine (5)”.
Response: corrected.
- In line 543, “N2” is necessary to be written “N2” and in line 558, “13C NMR” is necessary to be written “13C NMR”.
Response: corrected.
- In line 560, “C12H17DClN3OSi” is necessary to be corrected to “C12H18DClN3OSi”.
Response: corrected.
- In line 591, “C18H26ClN3OSi” is necessary to be corrected to “C18H27ClN3OSi”.
Response: corrected.
Reviewer 2 Report
In this manuscript the authors provide a thorought account on the directed lithiation of N-SEM protected 4-chloropyrrolopyrimidine and subsequent reactions of the C-6 lithiated intermediates with a series of oxo-compounds including enolazable ones. The degree of the lithiation was also monitored by quenching with D2O. The results of the appropriately designed and conducted research were clearly presented and interpreted. The manuscript is worth to be published, however prior to the final acceptance the following points need to be addressed.
- In order to check the efficiency of a sterically more demanding lithiating agent to avoid undesired C-4 amination at elevated temperature, at least one single control experiment with benzaldehyde should be performed using lithium tetramethylpiperidine instead of LDA.
- The formation of aldol condensation products was not detected when cyclic ketones were employed as electrophilic quenching components. This experimental finding should be reasoned in a few sentences.
- In Scheme 2. the structure of SEM should be corrected by deleting the terminal oxygen atom.
Author Response
- In order to check the efficiency of a sterically more demanding lithiating agent to avoid undesired C-4 amination at elevated temperature, at least one single control experiment with benzaldehyde should be performed using lithium tetramethylpiperidine instead of LDA.
Response: lithium tetramethylpiperidine was considered as a base in these reactions. However, due to the time limit for the revision we are not able to perform this reactions in a timely manner.
- The formation of aldol condensation products was not detected when cyclic ketones were employed as electrophilic quenching components. This experimental finding should be reasoned in a few sentences.
Response: We have included the following sentence: 1H NMR of the crude material in addition to 3p mostly contained 1 and cyclopentanone. There were no indications of aldol products being formed.
- In Scheme 2. the structure of SEM should be corrected by deleting the terminal oxygen atom.
Response: corrected
Reviewer 3 Report
The authors have investigated lithiation-addition at C-6 of protected 4-chloro-7H-pyrrolo[2,3-d]pyrimidine with different aldehydes and ketones. The addition of bis(2-dimethylaminoethyl) ether increased the yield.
16 compounds are obtained by reaction with aldehydes, cyclohexanone derivatives and acetophenones with yields in the range of 46-93%.
This paper is suitable to be published in Molecules after minor revision:
It is would be interesting to carried out the transformation of 3m to 8 reporting the corresponding yield.
IV in pag 3 line 91 should be in bold.
Pag 6, line 184, correct: p-methoxyacetohenone
Pag 7, line 218, correct: preaviusly
In Supporting Information: In Figure S10, the legend indicates 13C-NMR but the figure corresponds with 1H-NMR.
Author Response
It is would be interesting to carried out the transformation of 3m to 8 reporting the corresponding yield.
Response: A new sentence has been included, line 198: The dehydration can be affected under acidic conditions typically used to deprotect the SEM group, or by mesylation of the alcohol and treatment with a sterically hindered base as reported by Chaitanya et al. (data not shown).
IV in pag 3 line 91 should be in bold.
Response: corrected
Pag 6, line 184, correct: p-methoxyacetohenone
Response: corrected
Pag 7, line 218, correct: preaviusly
Response: corrected
In Supporting Information: In Figure S10, the legend indicates 13C-NMR but the figure corresponds with 1H-NMR.
Response: corrected
Reviewer 4 Report
Dear Authors:
I am very pleased to reviewed the article “Directed lithiation of protected 4-chloropyrrolopyrimidine: addition to aldehydes and ketones aided by bis(2-dimethylaminoethyl)ether” by Frithjof Bjornstad et.al. This article address a very relevant problem in the synthetic chemistry through C-6 position lithiation and their substitution with aldehyde and ketone for further fabrication. This manuscript is well written with it problem statement, justification, experimental and discussion with conclusion. I am happy to accept this manuscript to be published in the Journal Molecule.
Author Response
No issues were raised
Reviewer 5 Report
Please, see the attached file.

Author Response
For reader less aware of the field, addition of structures for additives TMEDA and BDMAE would be helpful.
Response: the structures has been inserted above Table 1.
In analytical work, the Authors used 1H-NMR to measure lithiation rate by quenching reactions with D2O and measuring then the deuterated and nondeuterated derivatives. is there any chance that relaxation times of deuterated and nondeuterated derivatives are remarkably different, and is this taken into account in NMR measurement parametrrs?
Response: This has not been evaluated. However, we expect this effect to be only minor. Further, no line broadening was observed which would be the case if this was a serious problem.
Also the quality of figure 1 was very low in the manuscript, probably by mistake.
Response: A new version of the figure has been inserted.
In the next part, the Authors have utilised the lithiated compound for reactions with aldehydes and ketones. The chemical space is well covered and the findings are discussed well and the reasoning is done well.
In the Scheme 3, the list of carbonyl compounds is missing letter i and acetophenone is marked to be j. This needs to be corrected and checked through the text.
Response: Scheme 3 has been corrected. There were no other mistakes in the text.
In lines 199-200 the Authors pripose derivative 3m for an intermediate of 8. However, this simple dehydration has not been carried out or mentioned to be reported in literature. This reaction would be valuable addition, either from the literature or done by the Authors.
Response: A new sentence has been included, line 198: The dehydration can be affected under acidic conditions typically used to deprotect the SEM group, or by mesylation of the alcohol and treatment with a sterically hindered base as reported by Chaitanya et al. (data not shown).